# A Retrospective Observational Study of Risk Factors for Denosumab-Related Osteonecrosis of the Jaw in Patients with Bone Metastases from Solid Cancers

**DOI:** 10.3390/cancers12051209

**Published:** 2020-05-12

**Authors:** Satoe Okuma, Yuhei Matsuda, Yoshiki Nariai, Masaaki Karino, Ritsuro Suzuki, Takahiro Kanno

**Affiliations:** 1Department of Oral and Maxillofacial Surgery, Shimane University Faculty of Medicine, Shimane 693-8501, Japan; okuma125@med.shimane-u.ac.jp (S.O.); yuhei@med.shimane-u.ac.jp (Y.M.); y.nariai@matsue-cityhospital.jp (Y.N.); karino71@med.shimane-u.ac.jp (M.K.); 2Department of Oral and Maxillofacial Surgery, Matsue City Hospital, Shimane 690-8509, Japan; 3Department of Oncology/Hematology and Innovative Cancer Center, Shimane University Hospital, Shimane 693-8501, Japan; rsuzuki@med.shimane-u.ac.jp

**Keywords:** denosumab-related osteonecrosis of the jaw, bone metastasis, denosumab, retrospective cohort study

## Abstract

This single-center retrospective observational study aimed to identify risk factors for developing denosumab-related osteonecrosis of the jaw (DRONJ) in stage IV solid cancer patients with bone metastases. In total, 123 consecutive patients who had received 120 mg of denosumab every 4 weeks at least twice between July 2014 and October 2018 were included. We surveyed their demographics, medical history, blood test, underlying disease, and intraoral findings. Fourteen patients (11.4%) developed DRONJ within a mean denosumab administration period of 4 months (range: 2–52 months). Univariate analyses showed a statistically significant correlation between DRONJ and hormone therapy, chemotherapy/molecular target drug, apical periodontitis, periodontal disease, sex and body mass index. Multivariate analysis showed a statistically significant correlation between DRONJ and hormone therapy (odds ratio [OR], 22.07; 95% confidence interval [CI], 2.86–170.24), chemotherapy and/or molecular targeted therapy (OR, 18.61; 95% CI, 2.54–136.27), and apical periodontitis (OR, 22.75; 95% CI, 3.20–161.73). These findings imply that collaborative oral examinations by oral specialists may reduce the risk of development of DRONJ in patients treated with denosumab for bone metastases from solid cancers.

## 1. Introduction

Bone metastasis is commonly found in patients with solid cancers [1]. The frequency of bone metastases is 65–75% in breast and prostate cancer, 40–60% in thyroid cancer, 30–40% in lung cancer, 40% in bladder cancer, 20–35% in renal cancer, 14–45% in malignant melanoma, and 5% in digestive organ cancer [2,3]. Bone metastasis, unlike the metastasis of other organs, is not life-threatening, but it does cause serious deterioration in the quality of life (QoL) because of symptoms such as pain, fractures, and paralysis [4].

Bone metastases are often treated with bisphosphonates (BPs), such as zoledronic acid, or the monoclonal antibody denosumab. Both treatments can prevent and alleviate the skeletal-related events (SREs) caused by bone metastasis from lung cancer, breast cancer, prostate cancer, and other cancers [5]. Moreover, these medicines improve the overall survival rate of patients with these cancers [6].

Denosumab is a human monoclonal antibody that binds to, and interferes with, the activation of the receptor activator of nuclear factor kappa-β ligand (RANKL), a potent stimulator of osteoclastogenesis [7]. RANKL functions as an anti-resorptive agent by inhibiting osteoclast function and the subsequent resorption of bone [8]. Denosumab suppresses osteoclastic cell formation and survival via the RANKL pathway and decreases SREs caused by bone metastases [9]. Unlike BPs, denosumab is effective in patients with renal dysfunction and helps to reduce the monthly dosing frequency [10]. The bioavailability of denosumab is high, ranging between 60% and 80%; the highest blood concentration is observed at 1–4 weeks after administration [11]. Denosumab inhibits osteoclast formation, function, and survival [5]. Furthermore, denosumab does not become embedded within bone tissue, and has a short half-life of 12.5 days; by comparison, BPs have a half-life of 10–12 years [12]. A double-blind phase III trial comparing denosumab and BPs showed that the overall survival, disease progression, and percentage of adverse events were similar among the groups. However, patients treated with denosumab exhibited a more delayed onset of SREs [13,14]. Therefore, denosumab is widely used as a substitute for BPs in patients with bone metastases, osteoporosis, and other bone diseases [13,14].

Anti-resorptive agents may cause osteonecrosis of the jaw (ONJ), called anti-resorptive agent-related osteonecrosis of the jaw (ARONJ,) and medication-related osteonecrosis of the jaw (MRONJ), which includes denosumab-related osteonecrosis of the jaw (DRONJ) and BP osteonecrosis of the jaw (BRONJ) [15]. The reported risk of developing denosumab-related ONJ (DRONJ) in patients with osteoporosis is 0.01–0.03%, while this risk is 1–3% for oncology patients, which was not significantly different from that in intravenous bisphosphonate users [15,16,17]. Initially, ONJ develops from exposed bone or bone probed by a periodontal probe through an intraoral or extraoral fistula. No surgical treatment is indicated for exposed or necrotic bone in patients who are asymptomatic and have no evidence of infection. However, surgical debridement/resection in combination with antibiotic therapy and local antimicrobial treatment is recommended for patients with exposed/necrotic bone, pain, infection, pathologic fractures, or extra-oral fistulae, together with conservative oral management and care [16,17]. Regardless of the disease stage, mobile segments of the sequestrum should be removed [18]. Moreover, a recent study indicated that segmental resection and immediate reconstruction with a reconstruction plate are recommended in severe cases to improve and maintain better oral and maxillofacial QoL [19].

ARONJ significantly affects QoL, and the decline in QoL is correlated with the ARONJ stage. The following factors found in ARONJ patients may contribute to decreased QoL: infected and painful necrotic jawbone; ulcerated, painful, and swollen oral mucosa; chronic sinus tract and facial disfigurement; impaired speech, swallowing, and eating; and frequent medical and dental collaborative evaluations and treatments [20,21,22].

Moreover, the surgical treatment of ONJ leads to poor QoL in cancer patients [23]. A recent systematic analysis of clinical trials reported that the overall incidence of DRONJ in patients with cancer was 1.7% [24]. The dominant risk factors for ARONJ are the cumulative dose and the number of administrations of BPs or denosumab [24]. Typically, ARONJ develops following a local infection or trauma to the bone or soft tissue. Poor oral hygiene, invasive procedures, such as tooth extraction or dental implant placement, and mucosal trauma from ill-fitting prostheses have been identified as risk factors for ARONJ [25]. Several other factors are thought to be associated with an increased risk of ARONJ, including the use of other cancer therapies or corticosteroids, smoking, and comorbidities such as anemia, diabetes mellitus, and renal failure [26,27,28].

However, the exact mechanisms underlying DRONJ remain unclear, and definitive treatment strategies have not yet been developed. Therefore, in this study, we retrospectively investigated the risk factors for DRONJ. The identification of such risk factors would facilitate the prediction and prevention of DRONJ onset, as well as improve the oral and maxillofacial QoL and treatment outcomes of patients with bone metastases from solid cancers.

## 2. Results

### 2.1. Patient Demographics and Characteristics

In total, 157 consecutive patients were enrolled in our study. Among them, 123 patients (57 males and 66 females) met the inclusion criteria, and their characteristics are shown in Table 1. The median age was 68.0 years, and the median BMI was 20.6. The performance status (PS) was 1 in 11 patients (8.9%), 2 in 51 patients (41.5%), 3 in 52 patients (42.3%), and 4 in 9 patients (7.3%). One hundred twenty-two patients (99.2%) received food orally. The median Brinkman index was 0. Ten patients (8.1%) consumed alcohol daily. In total, 16 patients (13.0%) had diabetes, 4 (3.3%) had rheumatoid arthritis, 60 (48.8%) had hypercalcemia, 4 (3.3%) had hypothyroidism, 7 (5.7%) had osteoporosis, 1 (0.8%) had vitamin deficiency, and 85 (69.1%) had anemia. Nineteen patients (15.4%) received antithrombotic therapy, while no patients were treated for osteomalacia, dialysis, or Paget’s disease. The median hemoglobin level was 11.4 g/dL, and those of total protein, albumin, cholesterol, calcium, and C-reactive protein were 6.7 g/dL, 3.4 g/dL, 180.0 mg/dL, 8.9 mg/dL, and 1.6 mg/dL, respectively.

The most common types of cancers in our cohort were breast (32 patients; 26.0%), lung (24 patients; 19.5%), prostate (16 patients; 13.0%), and colon cancer (13 patients; 10.6%). All patients had stage IV disease and bone metastases, and 67 patients (54.5%) had multiple metastases. Chemotherapy and/or molecular targeted therapy were given concurrently in 34 patients (27.6%), while 23 patients (18.7%) received hormone therapy. No patients received angiogenesis or tyrosine kinase inhibitors.

Intraoral evaluations and checkups were performed in all patients before the start of denosumab treatment. The median number of teeth was 18.0 (range: 0.0–32.0), and 48 patients (39.0%) used partial or complete dentures. Forty-four patients (35.8%) were diagnosed with apical periodontitis, with the presence of a root apical lesion shown by panoramic or dental X-rays. Fifty-four patients had periodontal disease with marginal periodontitis (43.9%), which were diagnosed by panoramic or dental X-rays, or based on a pocket depth of at least 4 mm.

The median follow-up period was 4 months (range: 2–52 months). Denosumab treatment was continued in 40 patients (32.5%), while in 2 patients (1.6%) it was discontinued. Eighty-one patients (65.9%) died during the follow-up period. Fourteen patients (11.4%) developed DRONJ, which had a median onset of 10 months after the start of denosumab treatment (range: 7–45 months). DRONJ was diagnosed in 11 of 32 patients (34.4%) with breast cancer, 2 of 16 patients (12.5%) with prostate cancer, and 1 of 13 patients (7.7%) with colon cancer. At the time of DRONJ diagnosis, three patients were in stage 1 (21.4%), five were in stage 2 (35.7%), and the remaining six were in stage 3 (42.9%) according to the AAOMS guidelines [18]. DRONJ had progressed rapidly in these patients within 8 weeks of the initial manifestation of maxillary or mandibular bone exposure. Images from a representative DRONJ case are shown in Figure 1.

### 2.2. Comparison of DRONJ and Non-DRONJ Patients

The demographics of DRONJ and non-DRONJ patients were compared (Table 2). Statistically significant differences were found in sex, height, Brinkman index, use of chemotherapy/targeted molecular therapy, use of hormone therapy, presence of apical periodontitis/root apical lesions and periodontal disease with marginal periodontitis, and the duration of denosumab treatment. No other variables differed significantly between the two groups.

### 2.3. Risk Factors for DRONJ

In univariate analyses, statistically significant predictors of DRONJ onset included hormone therapy (odds ratio [OR], 5.81; 95% confidence interval [CI], 1.80–18.81), chemotherapy/molecular target drug (OR, 4.26; 95% CI, 1.35–13.40), apical periodontitis (OR, 5.52; 95% CI, 1.62–18.84), periodontal disease (OR, 9.57; 95% CI, 2.04–44.91), sex (OR, 6.11; 95% CI, 1.31–28.60), and body mass index (OR, 1.18; 95% CI, 1.02–1.37) (Table 3). Furthermore, in multivariate analysis, statistically significant predictors of DRONJ onset included hormone therapy (OR, 22.07; 95% CI, 2.86–170.24), chemotherapy/molecular targeted therapy (OR, 18.61; 95% CI, 2.54–136.27), and apical periodontitis (OR, 22.75; 95% CI, 3.20–161.73) (Table 3).

## 3. Discussion

Bone metastases are a frequent complication of solid cancers, including breast cancer and prostate cancer, affecting 1.5 million patients worldwide [29]. Clinically important skeletal complications result from osteoclast-mediated bone destruction, often leading to severe pain, decreased QoL, instability, and neurological complications [30]. Bisphosphonates and denosumab have been used to alleviate SREs in cancer patients, and both reduce the risk of bone metastasis and SREs in colorectal, prostate, and breast cancers [30,31]. The overall survival rates of bone metastasis or lung cancer patients treated with denosumab are comparable or superior to that of patients treated with BPs [5,6,14,32].

In an integrated analysis of three major phase III head-to-head trials, denosumab was superior to BPs in preventing SREs [32]. However, in the clinic, irregular administration and discontinuation of denosumab therapy may affect its efficacy. The International Society for Pharmacoeconomics and Outcomes Research defines medication compliance as the act of conforming to the recommendations made by the provider with respect to the timing, dosage, and frequency of medication [33]. Medication persistence refers to the act of conforming to a recommendation that the patient continue treatment for the prescribed period of time [33]. 

Long-term use of denosumab could lead to several adverse effects, including ONJ, although a review study of the long-term treatment of osteoporosis found that denosumab treatment safely produced a continuous marked increase in bone mineral density at all body sites in women with postmenopausal osteoporosis [34]. However, based on concerns regarding adverse events related to putative RANKL inhibition or to bone turnover oversuppression, it is advised against longer-term administration [34]. Three major phase III trials reported that the incidence of DRONJ was low (cumulative incidence of only 1.8%; *n* = 52/2862) in patients receiving denosumab [35]. In the present study, DRONJ occurred 7–45 months after the start of therapy (median, 10 months). Similarly, a recent clinical study reported a mean onset time of 14 months (range: 8–25 months) after the administration of denosumab [15,36,37]. The median treatment duration for patients who did not develop DRONJ was 4 (range 2–52) months, primarily because of cancer deaths during the early part of the study. Therefore, prolonged continuous longer-term administration may have resulted in significantly more DRONJ cases in this study, because of the time-dependent nature of DRONJ development, considering its clinical impact on bone mineral density and bone turnover oversuppression [34,37].

DRONJ causes severe functional and masticatory disorders, and thus has a major impact on patient QoL [22]. Therefore, it is essential to identify patients at risk, limit the number of such cases, and establish protocols for early treatment. Boquete-Castro et al. analyzed data from seven randomized controlled trials on denosumab (including the adverse effects thereof), and found that the overall incidence of ONJ in patients with cancer who received denosumab was 1.7% (95% CI, 0.9–3.1%) [24]. In the German X-TREME study, 15 patients had suspected ONJ (1.3%) [30]. In a randomized controlled study involving the use of denosumab, 2% of breast cancer patients, 2.3% of prostate cancer patients, and 1.1% of patients with either a solid tumor or multiple myeloma developed ONJ [38]. In the present study, 14 patients (11.4%) with bone metastases from solid tumors developed DRONJ (median onset, 10 months). However, this study included only Japanese advanced-stage solid cancer patients with bone metastasis. The patients were older than those included in previous studies [24,30,38], and the median duration of denosumab administration was 4 months; a higher rate of DRONJ might have been observed with longer denosumab administration. The “Position Paper 2017 of the Japanese Allied Committee” revealed that the possible incidence of DRONJ in cancer patients could be higher than that in patients with osteoporosis. However, the incidence was only about 1.8% in a 3-year prospective follow-up study of patients treated with denosumab with breast, prostate, and other solid cancers or multiple myeloma; this possible incidence rate is based only on a foreign study [39]. Prospective studies of the incidence of ONJ have been conducted in cancer patients treated with zoledronic acid or denosumab [6,11]. Of 5723 patients with breast, prostate, and other solid cancers and multiple myeloma, 52 patients (1.8%) treated with denosumab and 37 patients (1.3%) treated with zoledronic acid (i.e., 89 cancer patients in total) developed ONJ in a 3-year follow-up [6,11]. It is unclear whether demographic factors were associated with DRONJ in this study; however, no such associations were described in previous studies [38,40]. As the vast majority of patients in these studies were white and ~75% were from the United States or Europe, differences in DRONJ incidence by race and geographic region remain unknown [40]. Further, no Japanese prospective study has been conducted in cancer patients with bone metastasis treated with denosumab. In this regard, the administration of denosumab oncology doses of 120 mg every 4 weeks for bone metastasis from solid cancers in Japan might trigger a higher incidence of DRONJ in Japanese.

Interestingly, the DRONJ rate and onset time after denosumab treatment in this study were consistent with those reported in a recent retrospective study investigating DRONJ in all cancer patients at any stage [37]. A single-center retrospective study performed in France reported DRONJ rates of 1% after 0–6 months of treatment and 8% after 30 months of treatment [37]. At 12 months of treatment, the rate reached 3%, but many patients were excluded because of missing or inaccurate data. Excluding patients previously treated with BPs also reduced the number of cases [37]. Therefore, we did not exclude patients who had previously received BPs. The current review study by the International Task Force on ONJ found that the incidence of ONJ in oncology patient populations is markedly higher than that in osteoporosis patient populations on denosumab [41].

The univariate analysis performed in this study showed that sex, BMI, use of chemotherapy and/or targeted therapy, use of hormone therapy, presence of apical periodontitis/root apical lesions and periodontal disease with marginal periodontitis, and the duration of denosumab treatment were significantly different between the non-DRONJ and DRONJ groups. Most DRONJ cases were female patients with breast cancer (11 of 14 DRONJ cases); this had a large effect on the sex distribution and might also affect BMI, as is often seen in studies where the majority of patients have breast cancer [37,40,41]. A recent review showed that tooth extraction was also associated with the development of DRONJ, and another retrospective study suggested tooth extraction, chemotherapy, poor oral hygiene, and ill-fitting dentures as possible risk factors for the development of DRONJ [42]. Other than the underlying advanced-stage solid cancers, systemic health conditions were not found to be risk factors for DRONJ [43]. Of note, while smoking and alcohol consumption were not risk factors for DRONJ, they were risk factors for oral diseases, including periodontal disease of marginal periodontitis and apical periodontitis [44]. Chemotherapy, targeted therapy, and hormone therapy modulate bone metabolism and might lead to immunosuppression. Further, the long-term use of systemic corticoids could also increase the risk as an immunosuppression factor, which may trigger the development of DRONJ [39,41]. In advanced-stage solid cancers, patients should be provided with evidence-based cancer care to improve QoL and life expectancy. However, in our study, no patients were treated with angiogenesis inhibitors or tyrosine kinase inhibitors, which might be significant risk factors for ONJ development. 

We also found associations of DRONJ with existing apical/periapical periodontitis and periodontitis. Extraction of symptomatic infected teeth or progressively worsening teeth, removal of bone edges, and complete mucoperiosteal wound closure were performed in all patients before starting denosumab treatment. Asymptomatic or well-maintained teeth with periapical periodontitis or marginal periodontitis were statistically significant risk factors. No patients developed DRONJ at tooth extraction sites because denosumab was administered after complete mucoepithelial closure. Furthermore, oral and dental examinations were performed regularly at local dental clinics and Matsue City Hospital. Interestingly, the number of remaining teeth did not differ significantly between the non-DRONJ and DRONJ groups. Of the 14 patients with DRONJ, 3 (21.4%) were symptomatic but stable in stage 1 after conservative management. The condition of five patients (35.7%) progressed to stage 2, and in the remaining six patients (42.9%) aggressive progression to stage 3 occurred, indicating that DRONJ patients require regular professional care from oral and dental specialists during and after denosumab treatment.

Multivariate regression analysis revealed statistically significant correlations between the onset of DRONJ and hormone therapy (OR, 22.07; 95% CI, 2.86–170.24), chemotherapy and/or molecular targeted therapy (OR, 18.61; 95% CI, 2.54–136.27) and apical periodontitis (OR, 22.75; 95% CI, 3.20–161.73). Thus, teeth with symptomatic, active apical periodontitis, as well as asymptomatic teeth with root apical lesions, should be carefully evaluated, treated, and potentially extracted before the start of denosumab treatment.

Further studies are required to elucidate the mechanisms underlying DRONJ development. Using an animal model of BRONJ and medication-related osteonecrosis of the jaw (MRONJ), Otto et al. showed that MRONJ developed in areas of gingival or periodontal infection [43]. Meanwhile, in a combined ligature-induced periodontitis/tooth extraction mouse model, a pre-existing inflammatory condition exacerbated MRONJ development after tooth extraction, performed following the administration of denosumab [42]. Several clinical studies have found that periodontal disease and apical periodontitis are risk factors for ONJ, similar to retrospective clinical studies on BRONJ [45]. Furthermore, Hallmer et al. reported a diverse range of bacteria (represented by 16 S rRNA sequences) in all MRONJ necrotic bone samples, and in 60% of visually healthy bone samples [46]. Eight dominant taxa were identified at the genus level, namely *Porphyromonas*, *Lactobacillus, Tannerella, Prevotella, Actinomyces, Treponema, Streptococcus*, and *Fusobacterium*. These anaerobic bacteria are commonly found under the gingival margin and root apical lesion in the oral cavity, and may be related to DRONJ [46]. According to recent studies, periodontal infection and immunosuppressive conditions in dentoalveolar regions might also cause DRONJ [27]. Denosumab inhibits RANKL by mimicking the effect of osteoprotegerin [47,48,49]. Given that RANKL is expressed in both T and B lymphocytes, denosumab may exert immunosuppressive effects. Nevertheless, as osteonecrosis does not occur in other areas of the body, the jaw is likely subject to higher bone turnover. Clinical trials are needed to determine why bone turnover is higher in the mandible, and why bone remodeling undergoes a change when RANKL inhibitors are administered [50]. 

Definitive DRONJ treatment strategies have not yet been established [39,41]. Patient characteristics including age, sex, disease status, DRONJ stage, lesion size, medication exposure, and medical and pharmacological comorbidities may all inform the treatment choice. However, specific mechanisms through which these factors influence the course of DRONJ and the response to treatment are largely unknown, such that individual treatment strategies rely on clinical judgment. Other important factors to consider are prognosis, life expectancy, QoL, and the patient’s ability to cope with DRONJ lesions. Unlike BPs, stopping denosumab treatment may facilitate prevention of DRONJ [41]. In some patients, discontinuation of denosumab treatment profoundly improved the oral condition and induced the separation of highly necrotic sequestra of the mandible. Although no data exist regarding the effectiveness of discontinuing denosumab, we suggest that denosumab therapy should be stopped when a diagnosis of DRONJ is confirmed because of the rapid progression thereof. Re-initiation of denosumab may be considered when disease progression or new bone-related symptoms occur.

We recommend that oncologists highlight the importance of maintaining good oral hygiene and recognizing early signs of DRONJ in the context of denosumab treatment for stage IV advanced cancer bone metastases. Furthermore, we recommend the administration of denosumab as part of the management and care of cases with bone metastases from solid cancers, in combination with chemotherapy, targeted therapy, or hormonal therapy. All patients should undergo oral and dental examinations before the start of denosumab treatment, and should receive regular oral examinations while receiving denosumab. A previous report showed that the incidence and severity of ONJ could be minimized through proactive education [42,45]. Therefore, a lower incidence, and higher rate of resolution, of DRONJ may be achieved by educating health care providers. It may also be helpful to encourage the patient’s oral specialist, regular dentist, and oral and maxillofacial surgeons to ensure that the patient understands the risks of cancer treatment.

This study had several limitations. First, it used a retrospective observational design. Second, the risk factors associated with denosumab mortality require further clarification. Furthermore, the characteristics of solid cancers associated with bone metastases, including tumor site, histopathological type, and genetic factors, should be further examined to identify risk factors for DRONJ. Moreover, for greater clinical relevance, the number of patients in future investigations should be increased, and multicenter studies should be performed. Finally, the median duration of denosumab treatment and follow-up period were short in our study, because the majority of patients (65.9%) died early, especially in the non-DRONJ group.

## 4. Patients and Methods

### 4.1. Data Sources and Search Strategy

This was a retrospective single-center observational study to evaluate risk factors for developing DRONJ. All patients received denosumab for bone metastasis of solid cancers between July 2014 and October 2018 at Matsue City Hospital, Shimane, Japan. The observation began on the first day of denosumab treatment. All patients underwent an oral and dental examination before denosumab treatment by the same two oral surgeons (S.O. and Y.N.) throughout the study period. Furthermore, all symptomatic infected or problematic teeth were extracted using minimally invasive techniques with complete mucoperiosteal closure. Denosumab treatment was started four weeks after complete mucoepithelial closure was obtained. Patients were also advised to receive regular oral and dental care throughout the denosumab treatment and were followed by the same two hospital oral surgeons every month in the department of oral and maxillofacial surgery on the date of denosumab administration. This study was conducted with the approval of the Medical Ethics Committee of Shimane University (No. 20181225-1) and the Ethics Committee of Matsue City Hospital (No. 2019A-0004).

### 4.2. Inclusion and Exclusion Criteria

The inclusion criteria were as follows: stage IV solid cancer with bone metastases, denosumab (RANMARK; Daiichi Sankyo Company, Ltd., Tokyo, Japan) administration by subcutaneous injection at a dose of 120 mg every 4 weeks at least twice, and denosumab dose adjustment based on the patient’s calcium level and renal function. The exclusion criteria were patients with missing or inaccurate data.

### 4.3. Study Variables

The surveyed items were as follows: patient characteristics (age, sex, height, weight, body mass index [BMI], reason for treatment, duration of treatment, withdrawal of treatment, Brinkman index, and alcohol consumption), medical history (diabetes, rheumatism, hypercalcemia, hypercalcemia, hypocalcemia, osteoporosis, osteomalacia, vitamin B deficiency, anemia, dialysis, Paget’s disease, and antithrombotic therapy), blood examination (hemoglobin, total protein, albumin, cholesterol, calcium, and C-reactive protein), underlying characteristics of the solid cancer (type of cancer, cancer stage, bone metastasis, multiple metastasis, concurrent chemotherapy and/or molecular targeted therapy, angiogenesis inhibitor, tyrosine kinase inhibitor, and hormone therapy), and intraoral findings (number of teeth, denture use, and apical periodontitis such as root apical lesions or periodontal disease with marginal periodontitis).

### 4.4. Study Outcomes

The DRONJ-affected site and staging were registered and included for DRONJ patients based on the guidelines of the American Association of Oral and Maxillofacial Surgeons (AAOMS) [18]. DRONJ was diagnosed if all of the following were present: current or previous treatment with antiresorptive or antiangiogenic agents, exposed bone or bone that could be probed through an intraoral or extraoral fistula in the maxillofacial region that had persisted for 8 weeks or more, and no history of radiation therapy to the jaw or obvious metastatic disease of the jaw.

### 4.5. Statistical Analyses

All statistical analyses were performed using SPSS version 26.0 software (IBM Japan, Tokyo, Japan). Background factors in the two groups were analyzed using the chi-squared and Mann–Whitney U-tests. Univariate and multivariate analyses of the risk factors for the development of DRONJ were conducted using logistic regression analysis. All study variables were selected using the stepwise method.

## 5. Conclusions

In conclusion, this retrospective observational study analysis revealed statistically significant correlations of DRONJ onset with hormone therapy, chemotherapy/molecular targeted therapy, and apical periodontitis. Close collaborative oral examination and regular maintenance of oral hygiene by oral specialists may reduce the incidence of DRONJ in cancer patients with bone metastases treated with denosumab.

## Figures and Tables

**Figure 1 cancers-12-01209-f001:**
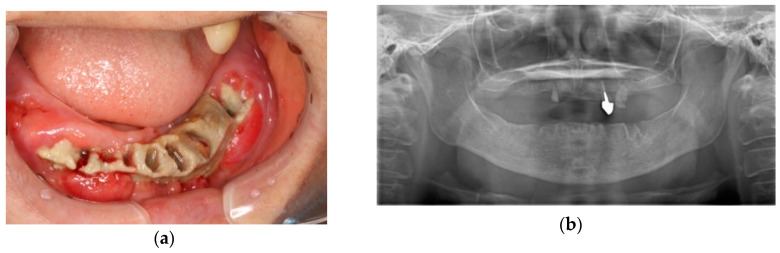
Representative case of a 61-year-old woman with bone metastases diagnosed as stage 3 denosumab-related osteonecrosis of the jaw (DRONJ) after surgical treatment and adjuvant hormone therapy for invasive ductal breast carcinoma. Denosumab was administered after chemotherapy and targeted therapy. After 16 months of denosumab administration, intraoral mandibular necrotic bone exposure, together with progressive spontaneous tooth loss was first observed during oral follow-up ((**a**) intraoral photo; (**b**) panoramic radiograph). Over the next 8 weeks, disease progression was aggressive, with surprisingly widespread necrotic bone exposure despite conservative management and care. After 18 months of denosumab administration, the patient was diagnosed with stage 3 DRONJ ((**c**) intraoral photo; (**d**) panoramic radiograph). The oncologist in charge recommended discontinuation of denosumab, while conservative treatment and regular care were continued. The patient was followed-up closely to check the progression of necrotic bone exposure, and complete separation of highly necrotic mandibular bone sequestra was confirmed after 10 months. The necrotic mandibular bone sequestra was surgically removed under local anesthesia at an outpatient clinic. Gradual mucoepithelial closure was observed, a newly fabricated denture was applied, and oral rehabilitation was achieved ((**e**) intraoral photo; (**f**) panoramic radiograph). Denosumab was not restarted after close consultation between the patient and oncologist. The patient remained systemically, locally, and intraorally stable, with gradual progression of multiple breast cancer metastases.

**Table 1 cancers-12-01209-t001:** Patient characteristics.

Description			*n* (%) or Median (Range)
Demographic factors	Sex	Male	57 (46.3)
		Female	66 (53.7)
	Age (years)		68.0 (25.0–95.0)
	Performance status	0	0 (0.0)
		1	11 (8.9)
		2	51 (41.5)
		3	52 (42.3)
		4	9 (7.3)
	Height (cm)		158.0 (130.0–178.0)
	Weight (kg)		51.4 (27.6–77.2)
	BMI ^1^		20.6 (13.3–31.0)
	Oral nutrition	Yes	122 (99.2)
	Brinkman index		0.0 (0.0–2460.0)
	Alcohol consumption	Yes	10 (8.1)
Medical history	Diabetes mellitus	Yes	16 (13.0)
	Rheumatoid arthritis	Yes	4 (3.3)
	Hypocalcemia	Yes	60 (48.8)
	Hypercalcemia	Yes	0 (0.0)
	Hypothyroidism	Yes	4 (3.3)
	Osteoporosis	Yes	7 (5.7)
	Vitamin B deficiency	Yes	1 (0.8)
	Anemia	Yes	85 (69.1)
	Antithrombotic therapy	Yes	19 (15.4)
Blood parameters	Hemoglobin (g/dL)		11.4 (6.3–17.7)
	Total protein (g/dL)		6.7 (4.1–8.4)
	Albumin (g/dL)		3.4 (1.5–4.6)
	Cholesterol (mg/dL)		180.0 (53.0–337.0)
	Calcium (mg/dL)		8.9 (6.2–10.2)
	C-reactive protein (mg/dL)		1.6 (0.0–23.7)
Underlying disease	Cancer type	Breast	32 (26.0)
		Lung	24 (19.5)
		Prostate	16 (13.0)
		Colon	13 (10.6)
		Pancreas	6 (4.9)
		Kidney	7 (5.7)
		Liver	8 (6.5)
		Uterus	2 (1.6)
		Stomach	6 (4.9)
		Bladder	3 (2.4)
		Other	6 (4.9)
	Bone metastasis	Yes	123 (100)
	Multiple metastases	Yes	67 (54.5)
	Chemotherapy and/or molecular targeted drug	Yes	34 (27.6)
	Angiogenesis inhibitor	Yes	0 (0.0)
	Tyrosine kinase inhibitor	Yes	0 (0.0)
	Hormonal therapy	Yes	23 (18.7)
Intraoral findings	Number of teeth		18.0 (0.0–32.0)
	Denture use	Yes	48 (39.0)
	Apical periodontitis	Yes	44 (35.8)
	Periodontal disease	Yes	54 (43.9)
Denosumab	Administration period (months)		4 (2–52)
	Reason for dropout	Continuing	40 (32.5)
		Cancelled	2 (1.6)
		Deceased	81 (65.9)
Follow-up (months)			4 (2–52)
DRON J ^2^	DRONJ ^2^ stage [18]	0	0 (0.0)
		1	3 (2.4)
		2	5 (4.1)
		3	6 (4.9)

^1^ body mass index, ^2^ denosumab-related osteonecrosis of jaw.

**Table 2 cancers-12-01209-t002:** Comparison between the DRONJ and non-DRONJ groups after follow-up.

Variables			*n* (%) or Median (Range)		
			Control (*n* = 109)	DRONJ (*n* = 14)	*p*-Value
Background factor	Sex	Male	55 (50.5)	2 (14.3)	0.011 *
		Female	54 (49.5)	12 (85.7)	
	Age (years)		68.0 (25.0–95.0)	64.0 (51.0–77.0)	0.155
	Performance status	0	0 (0.0)	0 (0.0)	0.422
		1	10 (9.2)	1 (7.1)	
		2	43 (39.4)	8 (57.1)	
		3	48 (44.0)	4 (28.6)	
		4	8 (7.3)	1 (7.1)	
	Height (cm)		160.0 (130.0–178.0)	152.0 (145.0–164.0)	0.010 *
	Weight (kg)		51.4 (27.6–77.2)	53.3 (31.0–72.5)	0.469
	BMI ^1^		20.4 (13.3–28.7)	21.7 (13.4–31.0)	0.055
	Oral nutrition	Yes	108 (99.1)	14 (100)	1.000
	Brinkman index		0.0 (0.0–2460.0)	0 (0.0–0.0)	0.001 **
	Alcohol consumption	Yes	10 (9.2)	0 (0.0)	0.602
Medical history	Diabetes mellitus	Yes	15 (13.8)	1 (7.1)	0.692
	Rheumatoid arthritis	Yes	3 (2.8)	1 (7.1)	0.387
	Hypocalcemia	Yes	55 (50.5)	5 (35.7)	0.397
	Hypercalcemia	Yes	0 (0.0)	0 (0.0)	-
	Hypothyroidism	Yes	4 (3.7)	0 (0.0)	1.000
	Osteoporosis	Yes	5 (4.6)	2 (14.3)	0.181
	Vitamin B deficiency	Yes	1 (0.9)	0 (0.0)	1.000
	Anemia	Yes	76 (69.7)	9 (64.3)	0.761
	Antithrombotic therapy	Yes	17 (15.6)	2 (14.3)	1.000
Blood examination	Hemoglobin (g/dL)		11.2 (6.3–17.7)	11.9 (8.3–15.9)	0.097
	Total protein (g/dL)		6.7 (4.1–8.4)	7.0 (5.6–7.8)	0.170
	Albumin (g/dL)		3.4 (1.5–4.6)	3.3 (2.6–4.5)	0.385
	Cholesterol (mg/dL)		179.0 (53.0–337.0)	201.5 (146.0–334.0)	0.232
	Calcium (mg/dL)		8.9 (6.2–10.0)	9.2 (6.4–10.2)	0.187
	C-reactive protein (mg/dL)		1.7 (0.0–23.7)	1.0 (0.0–4.7)	0.136
Underlying disease	Cancer type	Breast	21 (10.3)	11 (78.6)	-
		Lung	24 (22.0)	0 (0.0)	
		Prostate	14 (12.8)	2 (14.3)	
		Colon	12 (11.0)	1 (7.1)	
		Pancreatic	6 (5.5)	0 (0.0)	
		Kidney	7 (6.4)	0 (0.0)	
		Liver	8 (7.3)	0 (0.0)	
		Uterus	2 (1.8)	0 (0.0)	
		Stomach	6 (5.5)	0 (0.0)	
		Bladder	3 (2.8)	0 (0.0)	
		Other	6 (5.5)	0 (0.0)	
	Bone metastasis	Yes	109 (100)	14 (100)	-
	Multiple metastases	Yes	56 (51.4)	11 (78.6)	0.085
	Chemotherapy or molecular targeted drug	Yes	26 (23.9)	8 (57.1)	0.021 *
	Angiogenesis inhibitor	Yes	0 (0.0)	0 (0.0)	-
	Tyrosine kinaseinhibitor	Yes	0 (0.0)	0 (0.0)	-
	Hormonal therapy	Yes	16 (14.7)	7 (50.0)	0.005 **
Intraoral findings	Number of teeth		17.0 (0.0–32.0)	21.5 (0.0–32.0)	0.717
	Denture use	Yes	41 (37.6)	7 (50.0)	0.395
	Apical periodontitis	Yes	34 (31.2)	10 (71.4)	0.006 **
	Periodontal disease	Yes	42 (38.5)	12 (85.7)	0.001 **
Denosumab	Administration period (months)		4 (2–52)	10 (7–45)	0.001 **
	Drop out reason	Continuing	32 (29.4)	8 (57.1)	-
		Cancelled	1 (0.9)	1 (7.1)	
		Deceased	76 (69.7)	5 (35.7)	
DRONJ ^2^	DRONJ ^2^ stage [18]	0	0 (0.0)	0 (0.0)	-
		1	0 (0.0)	3 (21.4)	
		2	0 (0.0)	5 (35.7)	
		3	0 (0.0)	6 (42.9)	

^1^ body mass index, ^2^ denosumab-related osteonecrosis of jaw. **, *p* < 0.01; *, *p* < 0.05.

**Table 3 cancers-12-01209-t003:** Risk factors for DRONJ in multivariate analysis.

Variables	Univariate	Multivariate
Odds Ratio (CI)	Significance	Odds Ratio (CI)	Significance
Hormonal therapy	5.81 (1.80–18.81)	0.003	22.07 (2.86–170.24)	0.003
Chemotherapy/molecular target drug	4.26 (1.35–13.40)	0.013	18.61 (2.54–136.27)	0.004
Apical periodontitis	5.52 (1.62–18.84)	0.006	22.75 (3.20–161.73)	0.002
Periodontal disease	9.57 (2.04–44.91)	0.004		
Sex	6.11 (1.31–28.60)	0.022		
Body mass index	1.18 (1.02–1.37)	0.024

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
