# Peer review of "A Retrospective Observational Study of Risk Factors for Denosumab-Related Osteonecrosis of the Jaw in Patients with Bone Metastases from Solid Cancers"

_cancers, 2020, doi:10.3390/cancers12051209_

Round 1

Reviewer 1 Report

 [Reviewer(s)' Comments]
Reviewer: 1
Comments to the Author
This manuscript describes statistical significant correlation between DRONJ and hormone therapy, chemotherapy, apical periodontitis. Authors suggests that cooperative oral check by oral specialists reduce the risk of development of DRONJ in patients treated with denosumab for bone metastases from solid cancers, and interdisciplinary management and close involvement of patient’s oncologist is essential.

This article is very interesting and useful for oral surgeon and oncologist. Moreover, very refined analysis about DRONJ development had been performed in this article.

However, this article has some points to be improved for publication.

Major Points

1.Please minimize the inclusion of duplicate information between Introduction and Discussion section.

2.Please trim general contents in Discussion part, reader already thinks and knows .

3.Are there any difference between long-term (e.g more than 12 month ) group of denosumab and short-term group (e.g less than 12 months) of  denosumab in nature of denosumab?How do authors define long term use of denosumab?(L175〜183).Please refer following article.

Long-term treatment of osteoporosis: safety and efficacy appraisal of denosumab.

4.Delete redundant sentences in Discussion section. Readers of this article may be felt disturbed by the presence of abundant information. Please make the point clear.

5.Please elaborate on the incidence of ONJ comparing denosumab for malignant bone disease, and contrast this with denosumab's shorter half life, with using several references, not only single article (Ref12).

6.Fourteen 17 patients (11.4%) developed DRONJ within a mean denosumab administration period of 4 months 18 (range: 2–52 months).

We think that this rate is very high, compared to Position Paper 2017 of the Japanese Allied Committee.How do authors discuss the difference between this paper and position paper? Please refer following paper.

Antiresorptive agent-related osteonecrosis of the jaw: Position Paper 2017 of the Japanese Allied Committee on Osteonecrosis of the Jaw.

7.Please consider paragraph writing in this article. Paragraph structure should be clearly consisted of block that is divided into three sections: the topic sentence, the support sentence, and the concluding sentence.

8.How about the relationship between corticosteroids and incidence of DRONJ?

9.Please investigate association between comorbid conditions and DRONJ development. Systemic comorbidities are associated with medication‐related osteonecrosis of the jaws. Why don’t you investigate comorbidities score and polypharmacy score in development of DRONJ.

Patients with multiple systemic comorbidities and high levels of polypharmacy are more likely to develop MRONJ. CPS score is a effective tool to quantify the risk of MRONJ attributed to a patient's systemic condition. CPSs were grouped according to identified categories: mild (0–7 points), moderate (8–15 points), severe (15–21points) or morbid (>21 points).Please refer following article.

Systemic comorbidities are associated with medication‐related osteonecrosis of the jaws: Case–control study(Oral Dis. 2019 May;25(4):1107-1115. doi: 10.1111/odi.13046.)

Minor points

1.Please add CT image in  Figure 1 case.We are interested in separation image of sequestrum.

2.Denosumab has antiresorptive effects on bone and results in a rapid decrease in bone resorption and subsequent bone‐formation markers.Such changes have been reported for the bone‐resorption markers cross‐linked C‐telopeptide of type I collagen (CTX) in blood test.How about CTX value in blood test?

3.How do authors think relationship between DRONJ development body mass index?

4.Please add several papers other than Ref 32 in discussion section (line 168).

Author Response

Response to Comments/Suggestions from Reviewer 1

Dear reviewer 1

We are truly grateful to your critical comments and thoughtful suggestions on our manuscript. Based on these comments and suggestions, we have made careful modifications into our original manuscript. All changes made to the main text are in red. We here would like to say thank you for your helpful suggestions, which were very supportive for further improving this manuscript. You may kindly find our point-by-point responses to your comments/ questions as below.

Sincerely Yours

Professor. Takahiro Kanno, Corresponding Author for this article: cancers-794282

Comments to the Author

This manuscript describes statistical significant correlation between DRONJ and hormone therapy, chemotherapy, apical periodontitis. Authors suggests that cooperative oral check by oral specialists reduce the risk of development of DRONJ in patients treated with denosumab for bone metastases from solid cancers, and interdisciplinary management and close involvement of patient’s oncologist is essential.

This article is very interesting and useful for oral surgeon and oncologist. Moreover, very refined analysis about DRONJ development had been performed in this article.

However, this article has some points to be improved for publication.

Response: Thank you for your comments and suggestions, also, we really appreciate your kind recognition of this retrospective clinical work significance.

Major Points

  1. Please minimize the inclusion of duplicate information between Introduction and Discussion section.

Response: Thanks for your suggestions. We then minimized and deleted the duplicate information between Introduction and Discussion sections.

  1. Please trim general contents in Discussion part, reader already thinks and knows.

Response: Thanks for your suggestions. As related to the suggestion above, we then trimmed and deleted the duplicate information and general contents in Discussion section.

3.Are there any difference between long-term (e.g more than 12 month ) group of denosumab and short-term group (e.g less than 12 months) of denosumab in nature of denosumab?How do authors define long term use of denosumab?(L175〜183).Please refer following article.

Long-term treatment of osteoporosis: safety and efficacy appraisal of denosumab.

Response: Thanks for your suggestions. We appropriately refer the article as mentioned, and further revised this section of long-term administration of denosumab as in page 8, line 186-191 and line 195-200.

4.Delete redundant sentences in Discussion section. Readers of this article may be felt disturbed by the presence of abundant information. Please make the point clear.

Response: Thanks for your suggestions. As related to the suggestions above, we then trimmed and deleted the duplicate information, general contents and redundant sentences, and made discussion points clearer in Discussion section.

5.Please elaborate on the incidence of ONJ comparing denosumab for malignant bone disease, and contrast this with denosumab's shorter half life, with using several references, not only single article (Ref12).

Response: Thanks for your suggestions. According to your suggestion, we elaborated on the incidence of ONJ comparing denosumab for malignant bone disease using several references in Introduction section, as in page 2, line 60-62 with referring to appropriate references.

6.Fourteen 17 patients (11.4%) developed DRONJ within a mean denosumab administration period of 4 months 18 (range: 2–52 months).

We think that this rate is very high, compared to Position Paper 2017 of the Japanese Allied Committee. How do authors discuss the difference between this paper and position paper? Please refer following paper.

Antiresorptive agent-related osteonecrosis of the jaw: Position Paper 2017 of the Japanese Allied Committee on Osteonecrosis of the Jaw.

Response: Thanks for your helpful suggestions. This is a key discussion point. According to your suggestion, we referred that “Position Paper 2017 of the Japanese Allied Committee” and added the necessary discussions of the difference between this study and position paper referring to appropriate recent publications as references, as in page 9, line 210-212 and line 214-238.

7.Please consider paragraph writing in this article. Paragraph structure should be clearly consisted of block that is divided into three sections: the topic sentence, the support sentence, and the concluding sentence.

Response: Thanks for your suggestions. According to your suggestion, we re-structured each paragraph especially in Discussion section.

8.How about the relationship between corticosteroids and incidence of DRONJ?

Response: Thanks for your suggestions. In this study we didn’t investigate the administration of corticosteroids. We referred to another reference article to describe the possible relationship between corticosteroids and incidence of DRONJ in Discussion section according to your suggestion, as in page 9, line 253-254.

9.Please investigate association between comorbid conditions and DRONJ development. Systemic comorbidities are associated with medication‐related osteonecrosis of the jaws. Why don’t you investigate comorbidities score and polypharmacy score in development of DRONJ.

Patients with multiple systemic comorbidities and high levels of polypharmacy are more likely to develop MRONJ. CPS score is a effective tool to quantify the risk of MRONJ attributed to a patient's systemic condition. CPSs were grouped according to identified categories: mild (0–7 points), moderate (8–15 points), severe (15–21points) or morbid (>21 points).Please refer following article.

Systemic comorbidities are associated with medication‐related osteonecrosis of the jaws: Case–control study(Oral Dis. 2019 May;25(4):1107-1115. doi: 10.1111/odi.13046.)

Response: Thanks for your insightful suggestions. I’ve read through the article and found it profound merits and of great interest. However, in this study, we did retrospectively investigate the systemic and local factors which could be related to DRONJ development referring to the previous publications to start conducting this study based on this study protocol with approval of this study by IRB. Could you please let me have a chance of your proposal to further investigate comorbidities score and polypharmacy score, CPS score of an effective tool to quantify the risk of DRONJ in our next future study?

Minor points

1.Please add CT image in  Figure 1 case.We are interested in separation image of sequestrum.

Response: Thanks for your suggestions. However, we followed up the patient only for conservative oral health care management every month without taking CT and MRI, only with panoramic radiographs. The separation of large sequestrum in the mandible was easily diagnosed with panoramic radiographs and manifest clinically.

2.Denosumab has antiresorptive effects on bone and results in a rapid decrease in bone resorption and subsequent bone‐formation markers.Such changes have been reported for the bone‐resorption markers cross‐linked C‐telopeptide of type I collagen (CTX) in blood test.How about CTX value in blood test?

Response: Thanks for your suggestions. We fully agree with you that the CTX in blood serum examination could be one of the possible helpful markers to identify the boney turnover/resorption marker. However, we didn’t investigate that CTX in our institutes. Could you please let us plan further clinical research including your helpful suggestions?

3.How do authors think relationship between DRONJ development body mass index?

Response: Thanks for your suggestions. We then added the sentence to discuss this point with the DRONJ development, as in page 9, line 242-245.

4.Please add several papers other than Ref 32 in discussion section (line 168).

Response: Thanks for your suggestions. We then added relevant more reference articles to discuss this sentence, as in page 8, line 176-178.

Reviewer 2 Report

In the manuscript “cancers-794282” the authors report a “single-center retrospective observational study aimed to identify risk factors for developing denosumab-related osteonecrosis of the jaw (DRONJ) in stage IV solid cancer patients with bone metastases.” A total of 123 consecutive patients treated with denosumab (120 mg every 4 weeks at least twice between July 2014 and October 2018) were included in their study. “Fourteen patients (11.4%) developed DRONJ within a mean denosumab administration period of 4 months (range: 2–52 months). Univariable analyses showed a statistically significant correlation between DRONJ and hormone therapy, chemotherapy/molecular target drug, apical periodontitis, periodontal disease, sex and body mass index. Multivariate analysis showed a statistically significant correlation between DRONJ and hormone therapy……, chemotherapy and/or molecular targeted therapy……and apical periodontitis……” Based on these results, the authors concluded that “collaborative oral examinations by oral specialists may reduce the risk of development of DRONJ in patients treated with denosumab for bone metastases from solid cancers”.

This reviewer found the study of clinical interest.

Abstract

Please change “univariable” with univariate.

Introduction

Line 38: both treatments are also used to prevent SREs. Indeed, as reported by the authors “denosumab exhibited a more delayed onset of SREs” (line 53).

Lines 41-43: RANKL is a potent stimulator of osteoclastogenesis.

Line 44: Denosumab suppresses osteoclastogenesis and not osteoclastic activity.

Line 56: MRONJ is used in reference 15.

Lines 57-58: I wonder if this sentence is correct. According to AAOMS (reference 18 of the manuscript), “Patients may be considered to have MRONJ if all the following characteristics are present……Exposed bone or bone that can be probed through an intraoral or extraoral fistula in the maxillofacial region that has persisted for longer than 8 weeks……”.

Results

Line 88: why 34 patients were excluded ? Please see also Patients and Methods.

Lines 118-121: for this reviewer “according to AAOMS guidelines [18].” could be moved from line 121 to line 119.

Table 1

How the authors explain the absence of Hypercalcemia in their patients ?

Do the authors think that Osteomalacia, Dialysis and Paget’s disease are needed ? Please see also Table 2.

Please insert reference 18 for DRONJ staging.

Figure 1

Please change 2 with 1.

Was the diagnosis stage 3 DRONJ also done at 16 months ?

Table 2

Osteoporosis: were the 2 patients who developed DRONJ already under treatment with AR drugs?

Do the authors think that Osteomalacia, Dialysis and Paget’s disease are needed ? Please see also Table 1.

Table 3

Do not think the authors that Odds ratio (CI) and Significance for Periodontal disease, Sex and Body mass index (Multivariate) could be inserted as well ?

Can the authors explain the sentence “Periodontal disease, sex, body mass index, and administration period were adjusted for” ?

Discussion

Lines 165-166: do the authors mean progression, recurrence, metastasis and prevention for “risk of colorectal and breast cancers” ?

Lines 283-297: this section can be shortened. Many published papers deal with recommendations. In addition “A previous report” is devoid of reference.

Patients and Methods

Lines 321-325: why 34 patients were excluded ? The reasons of exclusion of these patients should be provided. Please see also Results.

Line 339: do the authors mean reference 18 ?

Conclusions

Are OR and CI values needed ? They have been already reported in Results and in Table 3.

Author Response

Response to Comments/Suggestions from Reviewer 2

Dear reviewer 2

We are truly grateful to your critical comments and thoughtful suggestions on our manuscript. Based on these comments and suggestions, we have made careful modifications into our original manuscript. All changes made to the main text are in red. We here would like to say thank you for your helpful suggestions, which were very supportive for further improving this manuscript. You may kindly find our point-by-point responses to your comments/ questions as below.

Sincerely yours

Professor. Takahiro Kanno, Corresponding Author for this article: cancers-794282

Comments to the Author

In the manuscript “cancers-794282” the authors report a “single-center retrospective observational study aimed to identify risk factors for developing denosumab-related osteonecrosis of the jaw (DRONJ) in stage IV solid cancer patients with bone metastases.” A total of 123 consecutive patients treated with denosumab (120 mg every 4 weeks at least twice between July 2014 and October 2018) were included in their study. “Fourteen patients (11.4%) developed DRONJ within a mean denosumab administration period of 4 months (range: 2–52 months). Univariable analyses showed a statistically significant correlation between DRONJ and hormone therapy, chemotherapy/molecular target drug, apical periodontitis, periodontal disease, sex and body mass index. Multivariate analysis showed a statistically significant correlation between DRONJ and hormone therapy……, chemotherapy and/or molecular targeted therapy……and apical periodontitis……” Based on these results, the authors concluded that “collaborative oral examinations by oral specialists may reduce the risk of development of DRONJ in patients treated with denosumab for bone metastases from solid cancers”.

This reviewer found the study of clinical interest.

Response: Thank you very much for your comments and suggestions, also, we really appreciate your kind recognition of this retrospective clinical work impact and significance.

Abstract

Please change “univariable” with univariate.

Response: Thanks for your suggestion. We then revised the sentence, as in page 1, line 19.

Introduction

Line 38: both treatments are also used to prevent SREs. Indeed, as reported by the authors “denosumab exhibited a more delayed onset of SREs” (line 53).

Response: Thanks for your suggestion. We then revised the sentence, as in page 1, line 38-40.

Lines 41-43: RANKL is a potent stimulator of osteoclastogenesis.

Response: Thanks for your suggestion. We then revised the sentence, as in page 1, line 42-43.

Line 44: Denosumab suppresses osteoclastogenesis and not osteoclastic activity.

Response: Thanks for your suggestion. We then revised the sentence, as in page 1, line 43-page 2, line 44 and in page 2, line 45-46.

Line 56: MRONJ is used in reference 15.

Response: Thanks for your suggestion. We then revised the sentence, as in page 2, line 58-59.

Lines 57-58: I wonder if this sentence is correct. According to AAOMS (reference 18 of the manuscript), “Patients may be considered to have MRONJ if all the following characteristics are present……Exposed bone or bone that can be probed through an intraoral or extraoral fistula in the maxillofacial region that has persisted for longer than 8 weeks……”.

Response: Thanks for your suggestion. This AAOMS position paper of reference 18 is summarized as in 4.4. Study outcomes in Patients and Methods section, as in page 12, line 363-364.

Results

Line 88: why 34 patients were excluded ? Please see also Patients and Methods.

Response: Thanks for your suggestion. We then revised the 4.2. Inclusion and exclusion criteria in Patients and Methods section, as related to another suggestion below, according to your suggestion, as in page 11, line 345-350.

Lines 118-121: for this reviewer “according to AAOMS guidelines [18].” could be moved from line 121 to line 119.

Response: Thanks for your suggestion. We then moved the sentence as suggested to do so, as in page 5, line 127.

Table 1

How the authors explain the absence of Hypercalcemia in their patients ?

Response: Thanks for your question. The absence of Hypercalcemia was diagnosed as that the calcium level was within the normal range in the blood examination.

Do the authors think that Osteomalacia, Dialysis and Paget’s disease are needed ? Please see also Table 2.

Response: Thanks for your suggestion. We then deleted these factors of Osteomalacia, Dialysis and Paget’s disease from both Table 1 and Table 2.

Please insert reference 18 for DRONJ staging.

Response: Thanks for your suggestion. We inserted the reference 18 for DRONJ staging in both Table 1 and Table 2, as suggested to do so.

Figure 1

Please change 2 with 1.

Response: Thanks for your suggestion. We revised the Figure 1 as suggested to do so.

Was the diagnosis stage 3 DRONJ also done at 16 months ?

Response: Thanks for your suggestion. The DRONJ diagnosis according to AAOMS diagnosis was done at 18 months of administration. Because the patient didn’t show any osteonecrosis at 15 months of follow-ups. These descriptions of Figure 1 are correct. After 16 months of denosumab administration, first, intraoral mandibular necrotic bone exposure together with progressive spontaneous tooth loss was observed. With following the next 8 weeks according to the AAOMS consensus, disease progression was aggressive, with surprisingly widespread necrotic bone exposure despite conservative management and care. Then, after 18 months of denosumab administration, the patient was the diagnosed with stage 3 DRONJ.

Table 2

Osteoporosis: were the 2 patients who developed DRONJ already under treatment with AR drugs?

Response: Thanks for your comment. Our 7 patients of osteoporosis had had oral BPs of AR drugs, not denosumab, as discussed in Discussion section, page 9, line 236.

Do the authors think that Osteomalacia, Dialysis and Paget’s disease are needed ? Please see also Table 1.

Response: Thanks for your suggestion. We then deleted these factors of Osteomalacia, Dialysis and Paget’s disease from both Table 1 and Table 2.

Table 3

Do not think the authors that Odds ratio (CI) and Significance for Periodontal disease, Sex and Body mass index (Multivariate) could be inserted as well ?

Response: Thanks for your insightful comment. We analyzed all the variable factors by using the stepwise selection method for both univariate and multivariate independently in an appropriate statistical manner of logistic regression analysis using SPSS ver. 26.0. This is because, there have been no prospective clinical study conducted so far to analyze the risk factor investigation of denosumab administration for DRONJ development in Japanese oncology patients with bone metastasis. Further, no definite or no reliable confounding factors of DRONJ have been evident especially in bone metastasis of Japanese oncology patients. We added the sentence in 4.5. Statistical analyses, as in page 12, line 373-374.

Other than this stepwise selection method for multivariate, we had also analyzed multivariate evaluation using the forced entry method of those 6 factors as significant variables of univariate analysis (Hormonal therapy, Chemotherapy / Molecular target drug, Apical periodontitis, Periodontal disease, Sex and Body mass index) as well. However, due to the coefficient of correlation; r=0.639 of Apical periodontitis and Periodontal disease and possibly multicollinearity of these. The results were as follows.

Univariate

Multivariate

Odds ratio (CI)

Significance

Odds ratio (CI)

Significance

Hormonal therapy

5.81 (1.80 – 18.81)

0.003

8.79 (1.57 – 49.20)

0.013

Chemotherapy / Molecular target drug

4.26 (1.35 – 13.40)

0.013

6.20 (1.27 – 30.22)

0.024

Apical periodontitis

5.52 (1.62 – 18.84)

0.006

3.57 (0.54 – 23.38)

0.184

Periodontal disease

9.57 (2.04 – 44.91)

0.004

4.45 (0.53 – 37.55)

0.170

Sex

6.11 (1.31 – 28.60)

0.022

3.05 (0.54 – 17.24)

0.207

Body mass index

1.18 (1.02 – 1.37)

0.024

1.04 (0.87 – 1.24)

0.700

Can the authors explain the sentence “Periodontal disease, sex, body mass index, and administration period were adjusted for” ?

Response: Thanks for your comment. I’m very sorry, but this was a completely wrong old sentence from our previous draft indifferent to this study article. We deleted this sentence.

Discussion

Lines 165-166: do the authors mean progression, recurrence, metastasis and prevention for “risk of colorectal and breast cancers” ?

Response: Thanks for your comment. This sentence was modified with adding the reference, not to be misunderstood, as in page 8, line 174-176.

Lines 283-297: this section can be shortened. Many published papers deal with recommendations. In addition “A previous report” is devoid of reference.

Response: Thanks for your comment. This paragraph was shortened and modified according to your suggestion, as in page 11, line 311-321.

Patients and Methods

Lines 321-325: why 34 patients were excluded ? The reasons of exclusion of these patients should be provided. Please see also Results.

Response: Thanks for your suggestion. We then revised the 4.2. Inclusion and exclusion criteria in Patients and Methods section, as related to the suggestion above, as in page 11, line 349-350.

Line 339: do the authors mean reference 18 ?

Response: Thanks for your suggestion. We then corrected the reference to 18, as in page 12, line 364.

Conclusions

Are OR and CI values needed ? They have been already reported in Results and in Table 3.

Response: Thanks for your suggestion. Yes, we then deleted these and revised Conclusion section, as in page 12, line 376-380.
